# Identification of Adult *Fasciola* spp. Using Matrix-Assisted Laser/Desorption Ionization Time-of-Flight (MALDI-TOF) Mass Spectrometry

**DOI:** 10.3390/microorganisms9010082

**Published:** 2020-12-31

**Authors:** Issa Sy, Lena Margardt, Emmanuel O. Ngbede, Mohammed I. Adah, Saheed T. Yusuf, Jennifer Keiser, Jacqueline Rehner, Jürg Utzinger, Sven Poppert, Sören L. Becker

**Affiliations:** 1Institute of Medical Microbiology and Hygiene, Saarland University, 66421 Homburg/Saar, Germany; issa.sy7@hotmail.com (I.S.); lena.margardt@uks.eu (L.M.); drngbede@hotmail.com (E.O.N.); jacqueline.rehner@uks.eu (J.R.); 2College of Veterinary Medicine, Federal University of Agriculture, Makurdi 970213, Nigeria; adahmi@yahoo.com; 3Kubwa Abattoir, Public Health Section, Department of Veterinary Services, Federal Capital Territory Administration, Abuja 901101, Nigeria; saheed.ty@gmail.com; 4Swiss Tropical and Public Health Institute, CH-4002 Basel, Switzerland; jennifer.keiser@swisstph.ch (J.K.); juerg.utzinger@swisstph.ch (J.U.); sven.poppert@swisstph.ch (S.P.); 5University of Basel, CH-4003 Basel, Switzerland

**Keywords:** diagnosis, *Fasciola gigantica*, *Fasciola hepatica*, fascioliasis, helminth, matrix-assisted laser/desorption ionization time-of-flight (MALDI-TOF) mass spectrometry, trematode

## Abstract

Fascioliasis is a neglected trematode infection caused by *Fasciola gigantica* and *Fasciola hepatica*. Routine diagnosis of fascioliasis relies on macroscopic identification of adult worms in liver tissue of slaughtered animals, and microscopic detection of eggs in fecal samples of animals and humans. However, the diagnostic accuracy of morphological techniques and stool microscopy is low. Molecular diagnostics (e.g., polymerase chain reaction (PCR)) are more reliable, but these techniques are not routinely available in clinical microbiology laboratories. Matrix-assisted laser/desorption ionization time-of-flight (MALDI-TOF) mass spectrometry (MS) is a widely-used technique for identification of bacteria and fungi; yet, standardized protocols and databases for parasite detection need to be developed. The purpose of this study was to develop and validate an in-house database for *Fasciola* species-specific identification. To achieve this goal, the posterior parts of seven adult *F. gigantica* and one adult *F. hepatica* were processed and subjected to MALDI-TOF MS to create main spectra profiles (MSPs). Repeatability and reproducibility tests were performed to develop the database. A principal component analysis revealed significant differences between the spectra of *F. gigantica* and *F. hepatica*. Subsequently, 78 *Fasciola* samples were analyzed by MALDI-TOF MS using the previously developed database, out of which 98.7% (n = 74) and 100% (n = 3) were correctly identified as *F. gigantica* and *F. hepatica*, respectively. Log score values ranged between 1.73 and 2.23, thus indicating a reliable identification. We conclude that MALDI-TOF MS can provide species-specific identification of medically relevant liver flukes.

## 1. Introduction

Fascioliasis is a food-borne parasitic disease in animals and humans, caused by the digenetic trematodes *Fasciola gigantica* and *Fasciola hepatica* [1]. While the occurrence of *F. gigantica* is restricted to areas of Asia and Africa [2,3], *F. hepatica* is widely distributed throughout Africa, Asia, Europe, Oceania and the Americas [4,5,6,7]. Fascioliasis and other food-borne trematode infections are classified as neglected tropical diseases. An estimated 2.4–17 million people are infected with *Fasciola* spp. [8]. However, these numbers may be considerable underestimations of the true number of infections, and hence, the global disease burden. In veterinary surveys, high prevalences of fascioliasis have been reported. For example, recent studies carried out in Lake Chad area and in a coastal region of Vietnam reported a prevalence in cattle of 68% and 23%, respectively; additionally, a significant risk for spillover of infections from animals to humans was noted [9,10].

The two *Fasciola* species share similar life cycles [1,11]. In brief, the life cycles are complex and require a snail intermediate host of the family Lymnaeidae (e.g., *Galba truncatula* for *F. hepatica*, *Lymnaea natalensis* for *F. gigantica* [12,13], and *Pseudosuccinea columella* for both species [14]). Adult worms are located in the biliary ducts of the definitive host (e.g., ruminants and humans). They release unembryonated eggs that are passed with the feces. These eggs become embryonated by contact with unprotected surface water. Subsequently, a miracidium hatches, which in turn infects the intermediate snail host. Free-swimming cercariae are released from the snail that form metacercariae and can attach to aquatic vegetation. Human infection occurs most frequently by the unintended ingestion of freshwater or vegetables (e.g., watercress) that are contaminated with infective metacercariae [7]. Following ingestion by humans or animals, immature worms excyst in the duodenum and penetrate the intestinal wall from where they migrate through the liver parenchyma [15].

Current methods used for the diagnosis of human and veterinary fascioliasis, in particular for the identification of adult worms, rely on morphological analyses of the trematodes [16], molecular methods (e.g., polymerase chain reaction (PCR)) and sequencing [7,11,17]. The latter two approaches have several limitations, including a lack of rigorous standardization of the morphological identification in different settings, relatively high costs and unavailability of PCR-based testing using specific primers for e.g., trematodes outside highly specialized research laboratories. While the morphological identification is a rapid and less costly procedure, *Fasciola* spp. are rarely detected outside endemic settings and the waning of parasitological experience among laboratory technicians in clinical laboratories is a particular challenge. Hence, the development of an accurate, rapid, less expensive and more accessible diagnostic technique for parasite identification would be desirable. During the past decade, matrix-assisted laser desorption/ionization time-of-flight (MALDI-TOF) mass spectrometry (MS) has become a widely employed tool for the diagnosis of bacteria and fungi in clinical samples [18,19,20]. More recently, a variety of studies have reported the use of MALDI-TOF MS as a potentially promising tool for the identification of mosquitoes [21,22], ticks [23], and, to a lesser extent, parasites (protozoa and helminths) [24]. In contrast to PCR-based techniques, MALDI-TOF MS allows for a broad, “untargeted” detection of microorganisms if these are present in the database used for pathogen identification. The goal of this study was to validate MALDI-TOF MS for the identification and differentiation of adult *Fasciola*.

## 2. Materials and Methods

### 2.1. Ethics Statement

For adult *Fasciola* samples originating from Nigeria, written permission was obtained from the management board of the Kubwa abattoir (Abuja, Nigeria) for collection and subsequent analyses of the fluke samples from slaughtered cattle (reference no.: FCT/ARD/TRN/004, 17 October 2019). Adult *F. hepatica* specimens were obtained from livers from a slaughterhouse in central Switzerland. Note that such specimens are utilized for routine diagnostic work and research purposes at the Swiss Tropical and Public Health Institute (Swiss TPH; Basel, Switzerland), adhering to local laws and regulations.

### 2.2. Sample Collection

For the current investigation, adult *Fasciola* worms were collected by experienced veterinarians from the liver tissue of slaughtered cattle at the Kubwa abattoir in Abuja, the Federal Capital Territory of Nigeria in December 2019 and January 2020. Samples were stored in 70% (*v*/*v*) ethanol at room temperature and were transferred to the Institute of Medical Microbiology and Hygiene in Homburg, Germany.

Additional samples were collected in early 2020 from the livers of cattle from a slaughterhouse in Oensingen, Switzerland. Livers were routinely screened by the slaughterhouse veterinarians. Infected livers were put aside and, on the same day, transferred and examined by experienced laboratory technicians at Swiss TPH. *Fasciola* specimens were stored in 0.7% (*v*/*v*) sodium chloride solution (NaCl) and transferred on ice to the Institute of Medical Microbiology and Hygiene in Homburg. Upon receipt in Homburg, samples were kept at −20 °C pending further examination.

### 2.3. Sample Preparation

Adult worms were removed from the storage solution and dried at room temperature to allow for evaporation of organic solvents. Two small pieces (each weighing approximately 15 mg) of the posterior part of each fluke were cut with a sterile scalpel to be used for subsequent molecular analyses and MALDI-TOF MS.

### 2.4. Molecular Analysis

#### 2.4.1. DNA Extraction, PCR and Sequencing

The DNeasy Blood and Tissue Kit (Qiagen GmbH; Hilden, Germany) was utilized for DNA extraction, adhering to the manufacturer’s instructions. Briefly, pieces measuring approximately 5 mm of each fluke were placed into a 1.5 mL Eppendorf tube, adding 280 µL lysis buffer and 20 µL proteinase K. The flukes were gently crushed in this mixture. Next, the mixture was incubated using a thermomixer (Eppendorf; Hamburg, Germany) at 56 °C and 800× *g* for 1 h. After digestion, a washing step was performed using AW1 and AW2 buffers from the Qiagen kits and a column with a silica membrane. The extracted DNA was eluted in 200 µL of AE buffer (Qiagen; Hilden, Germany) and stored at −20 °C pending molecular analyses.

PCR amplification of the partial mitochondrial cytochrome oxidase 1 gene (COX1) of all *Fasciola* specimens was carried out in Homburg, using a previously described protocol with the forward primer 5′-TTGGTTTTTTGGGCATCCT-3′ and the reverse primer 5′-AGGCCACCACCAAATAAAAGA-3′ [6]. The amplicons generated were sequenced using the Capillary Electrophoretic GenomeLab genetic analysis system (Beckman Coulter; Brea, CA, USA).

#### 2.4.2. Sequence Analysis and Species Identification

The forward and reverse sequences obtained were edited and combined to generate a consensus sequence for each specimen, using the software BioEdit^©^ version 7.2.5 (Tom Hall; Carlsbad, CA, USA) [25]. The consensus sequences were then queried against the National Center for Biotechnology Information (NCBI) GenBank database for identification, using the Basic Local Alignment Search Tool (BLASTn) [26]. Sequences obtained from the eight isolates that were used for MALDI-TOF MS database development during this study were submitted to GenBank, comprising the consecutive accession numbers MW258701 to MW258708.

### 2.5. MALDI-TOF Analysis

#### 2.5.1. Protein Extraction

A small piece corresponding to approximately 15 mg from the posterior part of each adult fluke was cut thinly with a sterile scalpel in order to facilitate the release of molecules from other locations than the fluke’s surface area, and was subsequently put into a 1.5 mL tube (Figure 1). Of note, no eggs can be found in the posterior part of *Fasciola*, as uterine structures are localized in the anterior part of this helminth. The “complete extraction protocol”, recommended by the manufacturer (Bruker Daltonics; Bremen, Germany) for protein extraction from bacteria for subsequent MALDI-TOF analysis, was readily adapted to the fluke samples. In brief, 300 µL of LC-MS grade water (Merck KG; Darmstadt, Germany) and 900 µL of 100% (*v*/*v*) ethanol (Merck KG) were added to the samples and mixed by vortexing. The mixture was centrifuged at 18,312× *g* for 2 min and the supernatant discarded. The pellet was resuspended in 50 µL of 70% formic acid and 50 µL of acetonitrile and mixed by vortexing. Of note, we also employed additional steps, including the use of Zirconium beads, during the development of our helminth extraction protocol, but did not observe differences in the obtained spectral profile. Hence, we decided not to use beads to reduce the working steps and the hands-on time per sample.

#### 2.5.2. Target Plate Preparation and Measurements

The protein extracts obtained with the mix of formic acid and acetonitrile above was centrifuged at 18,312× *g* for 2 min and 1 µL of the clear supernatant spotted onto the MALDI-TOF target plate, followed by overlaying with 1 µL of α-cyano-4-hydroxycinnamic acid (CHCA) matrix solution (Bruker Daltonics), composed of saturated CHCA 50% (*v*/*v*) of acetonitrile, 2.5% (*v*/*v*) of trifluoroacetic acid and 47.5% (*v*/*v*) of LC-MS grade water. Bacterial test standard (BTS) (Bruker Daltonics), which is an extract of *Escherichia coli* that is spiked with two high molecular weight proteins, was used to calibrate the machine. After drying at room temperature, the plate was placed into the Microflex LT Mass Spectrometer (Bruker Daltonics) for MALDI-TOF MS.

#### 2.5.3. MALDI-TOF MS Parameters

Measurements were performed using the AutoXecute algorithm in the FlexControl^©^ software version 3.4 (Bruker Daltonics). For each spot, 240 laser shots in six random positions were carried out automatically to generate protein mass profiles in linear positive ion mode with a laser frequency of 60 Hz, a high voltage of 20 kV and a pulsed ion extraction of 180 ns. Mass charge ratios range (*m*/*z*) were measured between 2 k and 20 k Da.

#### 2.5.4. Spectral Analysis and Database Creation

For the creation of species-specific main spectra profiles (MSPs), protein extracts of one *F. hepatica* and seven *F. gigantica* specimens were spotted onto the MALDI-TOF target plate eight times per sample. Next, each spot was measured four times to generate 32 raw spectra per sample. For each specimen, this procedure was carried out on two replicates on the same day (to demonstrate repeatability) and on one replicate on a different day (to demonstrate reproducibility). Hence, a total of 96 raw spectra were acquired for each sample, using the FlexControl^®^ software version 3.4 (Bruker Daltonics). These raw spectra were analyzed and curated using the FlexAnalysis^®^ software version 3.4 (Bruker Daltonics) for a “cleaning step”, i.e., to withdraw all flatlines and outlier peaks and to smoothen intensities and edit peak shifts within spectra whenever these exceeded 500 ppm. Following this editing step, replicates containing at least 22 remaining spectra were randomly chosen for the creation of species-specific MSPs. These MSPs were created using the automatic function of the MALDI Biotyper Compass Explorer^©^ software version 3.0 (Bruker Daltonics). Finally, the newly created MSPs of both *Fasciola* species were included in a previously developed in-house MALDI-TOF database for helminth identification, which already contained several nematodes (e.g., *Ascaris lumbricoides*), cestodes (e.g., *Taenia saginata*) and trematodes (e.g., *Schistosoma mansoni*).

MSP dendrogram analysis was carried out with the MALDI Biotyper Compass Explorer and employed the following parameters: distance correlation, linkage by average and score threshold values of 300 and 0 (arbitrary unit) for a single and a related organism, respectively. For a principal component analysis and a discriminatory analysis of the species-specific MSPs, we used the software BioNumerics^®^ version 7.5 (Applied Maths N.V.; Sint-Martens-Latem, Belgium).

#### 2.5.5. Validation Test

The newly developed in-house database was subjected to two different validation procedures. First, an internal validation, during which all raw spectra of *Fasciola* spp. obtained during the process of MSP creation were analyzed. Second, an external validation, during which raw spectra from new, unidentified *Fasciola* specimens were measured by MALDI-TOF MS to assess the database’s ability to reliably identify these samples.

All spectra were tested against the commercially available official database released by Bruker Daltonics for identification of bacteria and fungi (Bruker Taxonomy, Maldi Biotyper Compass Explorer version 3.0). Next, the spectra were subjected to a combination of the official database and our in-house helminth database. The reliability of identification was assessed by log score values (LSVs), which are generated for each result. We adhered to the “official” grading system put forth by the manufacturer for bacteria (i.e., LSVs may range between 0 and 3; LSV of ≥1.70 is considered as a threshold for identification; LSVs ranging between 1.70 and 1.99 indicate a reliable identification at the genus level; and LSVs equal to or higher than 2.0 are interpreted as reliable species identification).

## 3. Results

### 3.1. Comparative Analysis of Samples Used for MSP Database Creation

For the creation of MSPs to be included in the in-house database, we included seven randomly selected adult flukes obtained from cattle in Nigeria, which had previously been identified as *F. gigantica* based on morphological characteristics, and one adult *F. hepatica* provided from Switzerland. Partial sequencing of the COX1 confirmed the species diagnosis in the eight reference samples. In comparison to previously deposited sequences, the BLAST analysis revealed sequence homologies between 98.5% and 99.8% for *F. gigantica* (reference accession numbers: MN586868.1, MN586869.1, MN913872.1 and MN913873.1) and 99.4% for *F. hepatica* (reference accession number: GQ231551.1). The visualization of spectra obtained by MALDI-TOF MS is displayed in Figure 1. Considerable differences between *F. gigantica* and *F. hepatica* were observed (Figure 2A), which were confirmed by a dendrogram analysis (Figure 2B). Of note, there was some heterogeneity with regard to the spectra within the *F. gigantica* cluster (e.g., samples FGN23 and FGN24), which was further substantiated by a principal component analysis and discriminant analysis (Figure 3). The COX1 sequences of these two samples (FGN23 and FGN24) showed sequence homologies of 99.5% and 98.6%, respectively, to the GenBank reference sequences of *F. gigantica*.

### 3.2. Internal Database Validation

When the reference spectra were analyzed with the commercially available database for bacteria and fungi identification, no reliable identification was achieved, and all LSVs were below 1.7. In the subsequent analysis of the raw spectra obtained during MSP creation, which used a combination of the official and the in-house database, a reliable species identification as either *F. gigantica* or *F. hepatica* was achieved in all eight samples, with LSVs ranging from 2.17 to 2.86 (average LSV = 2.5).

### 3.3. Analysis of Samples for External Database Validation

A total of 78 *Fasciola* samples were subjected to MALDI-TOF analysis, using the newly developed database. There were 75 samples from Nigeria that had previously been identified as *F. gigantica*, based on morphological characteristics. The remaining three specimens stemmed from Switzerland and were reported as *F. hepatica*. PCR was performed on all samples and confirmed the species identification in all cases.

When using the previously developed in-house database and considering an LSV threshold of ≥1.70, MALDI-TOF MS correctly identified 74/75 (98.7%) of the *F. gigantica* and 3/3 (100%) of the *F. hepatica* samples. In one sample, MALDI-TOF MS did not identify sufficient protein spectra to provide an identification. When an LSV threshold of ≥2.0 was used, the identification rate was 31/75 (41.3%) for *F. gigantica*, while it remained unchanged for *F. hepatica* (Table 1). Of note, no *F. gigantica* sample was misidentified as *F. hepatica* and vice versa. 

## 4. Discussion

The purpose of this study was to determine whether MALDI-TOF MS can be utilized as a diagnostic tool for the identification and differentiation of adult *Fasciola* species. Our results show that MSPs created from repetitive measurements of eight adult *Fasciola* specimens, utilized as part of a helminth-specific in-house database, allowed for an unambiguous identification of 78 additional *Fasciola* samples. Indeed, 77 of 78 specimens were correctly identified, and there was no misidentification at the species level. A deeper investigation using principal component analysis and discriminant analysis for comparative spectra visualization confirmed species-specific differences that enable accurate diagnosis. In the case of the single specimen that was not correctly identified, we had already noted a slightly different morphological aspect on macroscopic inspection, and no meaningful protein spectra of sufficient quality were obtained by MALDI-TOF MS, which probably indicates previous degradation of the proteins.

MALDI-TOF MS has become a standard diagnostic tool for bacteria [27,28] in the microbiological laboratory and is now increasingly being utilized for identification of yeasts and filamentous fungi as well as mycobacteria. In research settings, entomological studies have also employed MALDI-TOF for identification of ticks [23], mosquitoes [21,29], fleas [30], and lice [31]. Yet, data on application of MALDI-TOF MS for parasites, particularly helminths, are scarce. A recent systematic review identified only five published studies pertaining to diagnostic helminthology [24], using MALDI-TOF MS specifically for *Ascaris* spp., cyathostomin helminths, *Dirofilaria* spp. and *Trichinella* spp. Recently, two studies described the application of MALDI-TOF MS for identification of helminths; one focusing on *Trichinella* spp. in France [32] and the other on *Anisakis* spp. in Italy [33]. Using different cestode, nematode and trematode samples, our group has also generated species-specific MSPs and developed an in-house database for helminth identification, which will prospectively be validated on well characterized clinical samples [34]. It is important to note that the protocols used for protein extraction and MALDI-TOF spectra acquisition varied slightly between studies, and the development of one standardized approach will be an important feature to generate (i) accurate databases; and (ii) specific and reproducible results across different laboratories in future studies.

Interestingly, while creating the MSPs for the in-house database, it was observed that two out of seven *F. gigantica* samples used (i.e., isolates no. FGN23 and FGN24) clustered together in a group with slightly different spectral patterns than the other five *F. gigantica* samples. Such observed intra-species differences in the mass spectra profiles could be explained by a minor genetic variation (e.g., a non-synonymous mutation, which could affect the protein profile). In this context, it is important to mention that several studies have reported the existence of an intermediate “hybrid” species of *Fasciola*, which can only be discriminated by specific molecular methods [34]. Liu and colleagues [35] studied the sequence data of protein-encoding genes and showed that the intermediate form of *Fasciola* is more closely related to *F. gigantica* than *F. hepatica*. Similar studies from sub-Saharan Africa have suggested that the epidemiology of fascioliasis in this part of the world may be more complex than previously thought, and that *F. gigantica* is not the only species occurring [36]. “Hybrid” *Fasciola* spp. have also been reported from Chad [37], and these might also occur in Nigeria, where the *F. gigantica* samples for the current MALDI-TOF MS were obtained. Since “hybrid” species cannot be accurately identified by amplification of the COX1 as performed here, further molecular investigations pertaining to the genomics (e.g., sequencing of the internal transcribed spacer (ITS) region as reported by Evack and colleagues [37]) of these samples will be interesting. Indeed, while we aligned and comparatively analyzed the obtained COX1 sequences, we were unable to identify specific variable sites, which would have allowed to accurately identify “hybrid” isolates.

Our study is limited by the small sample size for *F. hepatica* (i.e., only four adult worms), analysis of samples stemming from only two different geographical areas (i.e., Nigeria and Switzerland) and the use of samples preserved in different media (70% ethanol and 0.7% NaCl for *F. gigantica* and *F. hepatica*, respectively). It is important to mention that potential effects of different preservation media used were not thoroughly investigated in this proof-of-concept study. Another limitation relates to the hosts, as we only analyzed *Fasciola* spp. stemming from animals and further research is needed to confirm that our developed in-house database would also reliably identify flukes from other hosts, e.g., specimens extracted from human bile ducts. It should also be noted that more than half of all *F. gigantica* samples were identified with LSVs < 2.0, which would only relate to a genus-specific, but not to a species-specific identification if thresholds for bacteria were used. However, MALDI-TOF MS did not misidentify any of the two *Fasciola* species. For the further development and improvement of a helminth-specific MALDI-TOF database, a more comprehensive investigation with specimens from other geographical settings in Africa, Asia, Europe and the Americas and samples originating from different hosts would be desirable. Additionally, while we only analyzed adult flukes, future studies should also consider *Fasciola* eggs and different larval stages to obtain a more accurate taxonomic typing and to further improve the species identification via MALDI-TOF MS. Indeed, a direct MALDI-TOF-based identification of *Fasciola* eggs in stool samples of human and veterinary origin would represent a major achievement to improve the diagnosis of fascioliasis, including “hybrid” species, in many laboratories. Pending further research on the effect of different storage solutions on the resulting protein spectra, this method might also be employed as a suitable alternative to PCR for a diagnostic identification of helminths after storage for many years (e.g., historical parasite collections in research laboratories).

## 5. Conclusions

We conclude that MALDI-TOF MS is a promising tool for rapid and reliable identification of adult *Fasciola* and potentially other food-borne trematode species. It is important to note that the creation and validation of a specific in-house database is necessary for identification of taxonomic groups (e.g., food-borne trematodes) that are not covered by commercial databases. To our knowledge, this study is the first to employ MALDI-TOF for differentiation of the causative agents of fascioliasis.

## Figures and Tables

**Figure 1 microorganisms-09-00082-f001:**
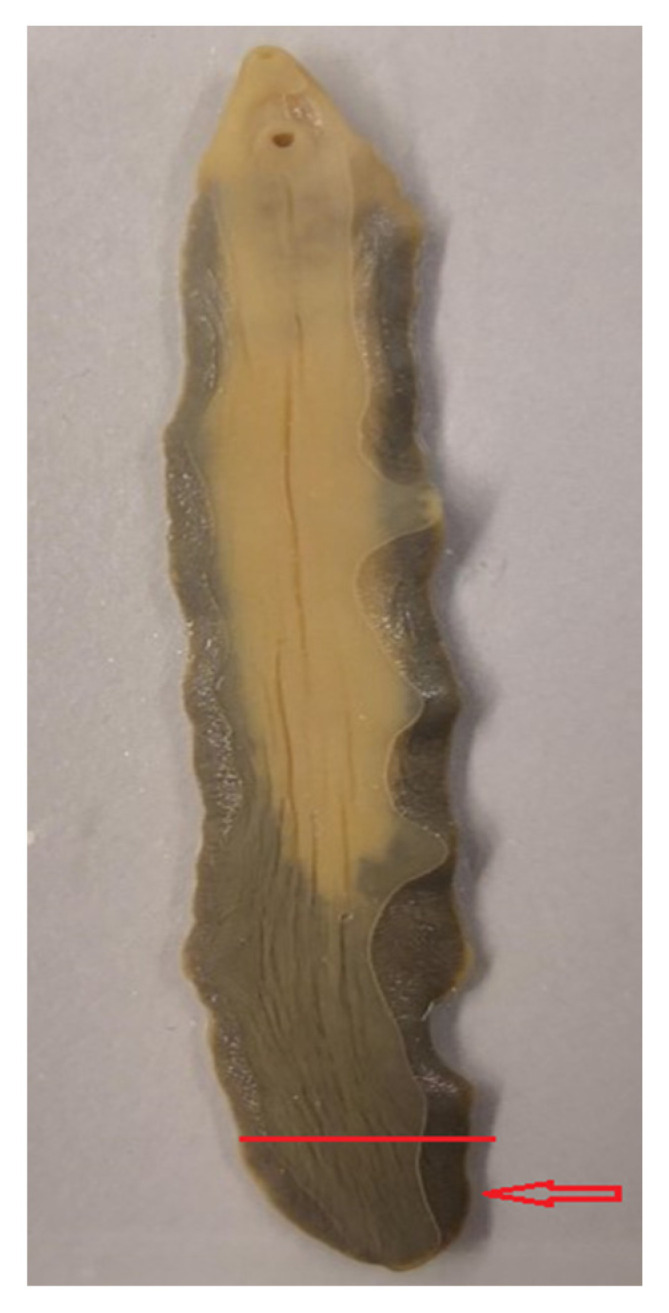
Morphology of an adult *Fasciola gigantica* fluke. The red line shows the area where the incision of the helminth was performed, and the red arrow indicates the posterior body part that was used for molecular analysis and MALDI-TOF MS.

**Figure 2 microorganisms-09-00082-f002:**
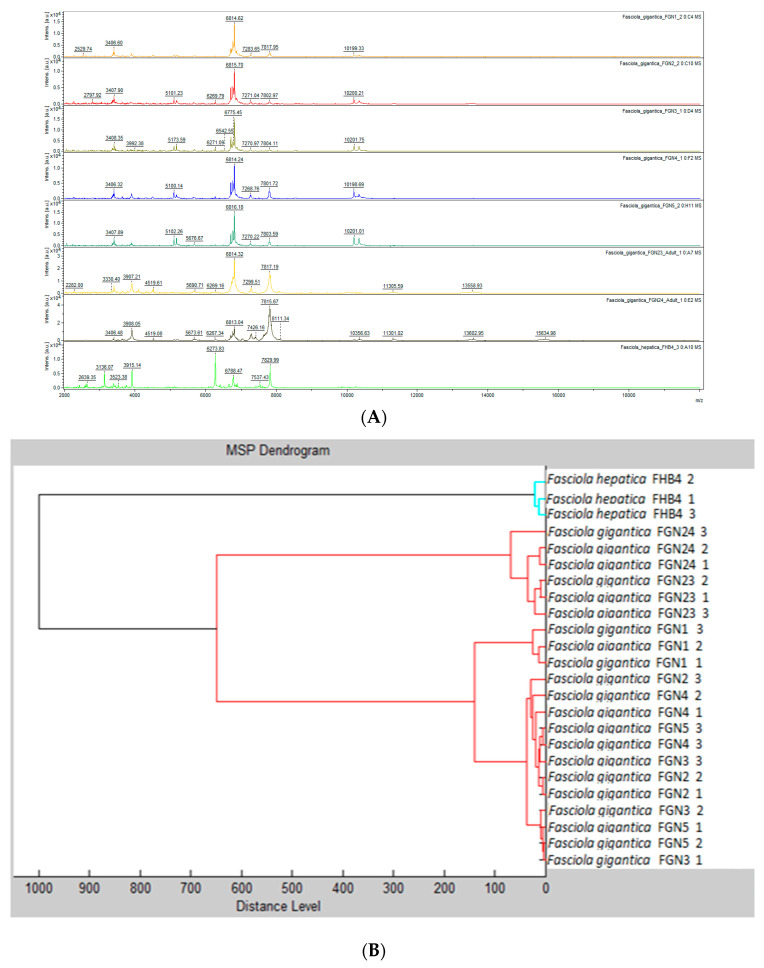
(**A**) Spectra obtained by matrix-assisted laser/desorption ionization time-of-flight (MALDI-TOF) mass spectrometry (MS) for seven adult *Fasciola gigantica* (FG) specimens and one adult *Fasciola hepatica* (FH) specimen. The peak intensities of ionised molecules are shown on the y-axis and the corresponding mass on the x-axis. Numbers indicate the resulting mass-to-charge-ratios. (**B**) Dendrogram analysis displaying the relatedness of the different samples. Of note, each specimen was measured thrice to assure reproducibility.

**Figure 3 microorganisms-09-00082-f003:**
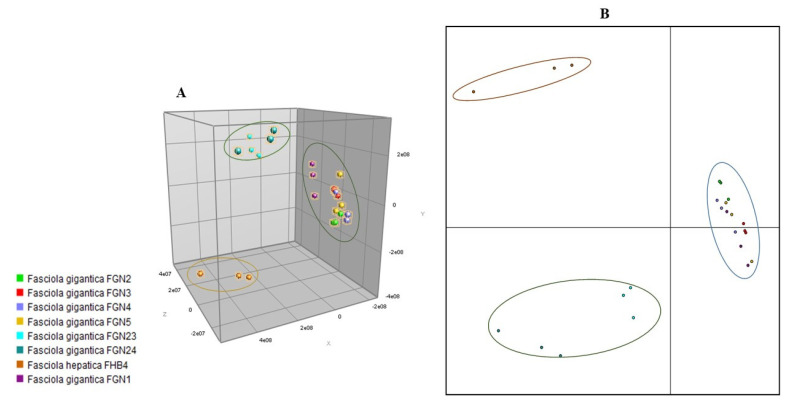
Statistical analysis of main spectra profiles (MSPs) obtained during the creation of an in-house database for identification of adult *Fasciola gigantica* (FG) and *Fasciola hepatica* (FH) using MALDI-TOF MS. (**A**) Three-dimensional view of a principal component analysis, displaying three distinct clusters, i.e., (i) *F. hepatica* FHB4; (ii) *F. gigantica* FGN23 and FGN24; and (iii) the remaining *F. gigantica* samples. (**B**) Two-dimensional view of a discriminant analysis, using the same samples. The analyses were carried out using the software BioNumerics.

**Table 1 microorganisms-09-00082-t001:** Identification of 78 adult *Fasciola* spp. samples obtained from Nigeria and Switzerland by MALDI-TOF MS, using a newly developed in-house database. LSV, log-score value.

Species	Number of Samples	Identification		LSV Range
		LSV ≥ 1.70	LSV ≥ 2.00	
*Fasciola gigantica*	75	74 (98.7%)	31 (41.3%)	1.73–2.13
*Fasciola hepatica*	3	3 (100%)	3 (100%)	2.15–2.23

## Data Availability

The data presented in this study are available on request from the corresponding author.

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
