# Peer review of "Identification of Adult Fasciola spp. Using Matrix-Assisted Laser/Desorption Ionization Time-of-Flight (MALDI-TOF) Mass Spectrometry"

_microorganisms, 2020, doi:10.3390/microorganisms9010082_

Round 1

Reviewer 1 Report

The use of MADLI for identification of parasites is an interesting idea and one that warrants further investigation. The overall approach of library entry creation was sound. I found the finding and clustering of spectral signals interesting and something that could be built upon in the future.

I have a few concerns that need to be addressed. The quantity of material processed for MALDI can impact the quality of results. The phrase "small pieces" is subjective. A reference for the volume or the mass of the material is needed to be reproducible by others. In addition, for a multicellular organism there may be differences in MALDI signature depending on where the sample was collected. A graphic or image indicating sample location is needed in place of simply "posterior end".

The processing of the sample could also be more clear. When the dried piece of posterior worm segment was mixed with the extraction reagents, did it dissolve or break up during processing? Several MALDI protocols involve use of bead beating for proper release of molecules. Why was this not needed? How do we know that the processing appropriately solubilized the components for MALDI? If the segment did not break up, or is just surface skin, could the signature produced be from surface bacteria that washed off?

The overall summary of the article also missed a key point in their findings. This research has a short coming in that routine ID to the species level was not possible for the majority of Fasciola gigantica assessed. The authors should more thoroughly address this short coming of the technology. There is little clinical value in distinguishing between Fasciola species. Most diagnostic labs would not ID beyond the genus, MALDI does not need to be better than that since it will not impact treatment decisions.

A minor suggestion is a missed opportunity to justify this study. Morphological ID is fast and cheap. This kind of technology is most likely to be of value in diagnostic assistance because these organisms are rare and it is difficult to maintain trained, competent staff to do traditional morphologic identification. This technology would make ID more accessible to a broader range of diagnostic labs.

On line 21 the authors reference "slaughtered animals" as a source of these flukes but their paper proposes use of this technology in people. I recommend mentioning that adult flukes can come from humans as well. In the discussion it would also be good to mention that a potential short coming of this study is that it is based on flukes collected from animals only. It is not clear that the MALDI signal would be the same for flukes grown in humans. This limits the appropriate use of the new MALDI library to animal grown flukes until further validated is performed.

There was an inconsistency on line 282 that references "4 adult worms", this is not consistent with the rest of the text.

In summary, the methods would not be easily reproducible based on this text alone. Additional information is needed before it is acceptable for publication. With the proposed revisions I can see this information being a useful contribution to the field.

Author Response

Reviewer #1:

The use of MALDI for identification of parasites is an interesting idea and one that warrants further investigation. The overall approach of library entry creation was sound. I found the finding and clustering of spectral signals interesting and something that could be built upon in the future.

Response: We thank Reviewer #1 for the general interest in our work and the very helpful suggestions, which we addressed to our level best while revising our manuscript.

I have a few concerns that need to be addressed. The quantity of material processed for MALDI can impact the quality of results. The phrase "small pieces" is subjective. A reference for the volume or the mass of the material is needed to be reproducible by others. In addition, for a multicellular organism there may be differences in MALDI signature depending on where the sample was collected. A graphic or image indicating sample location is needed in place of simply "posterior end".

Response: We thank Reviewer #1 for these suggestions. We now provide a more detailed description of the quantity used for our investigation, as follows: “Two small pieces (each weighing approximately 15 mg) of the posterior part of each fluke were cut with a sterile scalpel to be used for subsequent molecular analyses and MALDI-TOF MS” (see revised manuscript, lines 103-105). Additionally, we have also included a photo of an adult fluke on which the sampling area is clearly indicated by a red arrow (see revised manuscript, new Figure 1).

The processing of the sample could also be more clear. When the dried piece of posterior worm segment was mixed with the extraction reagents, did it dissolve or break up during processing? Several MALDI protocols involve use of bead beating for proper release of molecules. Why was this not needed? How do we know that the processing appropriately solubilized the components for MALDI? If the segment did not break up, or is just surface skin, could the signature produced be from surface bacteria that washed off?

Response: While revising our manuscript, we tried to incorporate these additional requests to enhance clarity. First, the dried part of the adult fluke was not entirely dissolved during the sample processing, but it was cut thinly to allow for release of proteins and molecules from other locations than the surface of the fluke (see revised manuscript, lines 131-133). Second, during the validation process for helminth samples to be subjected to MALDI-TOF MS, we also utilised different extraction protocols, e.g. with and without the use of Zirconium beads. Yet, we did not observe a difference in the spectral profile and thus decided not to use beads to reduce the working steps and the hands-on time per sample (see revised manuscript, lines 141-144). Third, to ensure that our measured mass spectra profiles did not stem from e.g. surface bacteria, we submitted them for analysis to the official database by Bruker Daltonics for bacterial identification, i.e. “All spectra were tested against the commercially available official database released by Bruker Daltonics. […] When the reference spectra were analysed with the commercially available database for bacteria and fungi identification, no reliable identification was achieved, and all LSVs were below 1.7” (see revised manuscript, lines 191-192 and lines 230-231).

The overall summary of the article also missed a key point in their findings. This research has a short coming in that routine ID to the species level was not possible for the majority of Fasciola gigantica assessed. The authors should more thoroughly address this short coming of the technology. There is little clinical value in distinguishing between Fasciola species. Most diagnostic labs would not ID beyond the genus, MALDI does not need to be better than that since it will not impact treatment decisions.

Response: We are grateful to Reviewer #1 for this important comment, which we addressed in the ‘Discussion’ as follows: “It should also be noted that more than half of all F. gigantica samples were identified with LSVs <2.0, which would only relate to a genus-specific, but not to a species-specific identification if thresholds for bacteria were used” (see revised manuscript, lines 303-305).

A minor suggestion is a missed opportunity to justify this study. Morphological ID is fast and cheap. This kind of technology is most likely to be of value in diagnostic assistance because these organisms are rare, and it is difficult to maintain trained, competent staff to do traditional morphologic identification. This technology would make ID more accessible to a broader range of diagnostic labs.

Response: This issue has been addressed as follows: “While the morphological identification is a fast and less costly procedure, Fasciola spp. are rarely detected outside endemic settings and the waning of parasitological experience of laboratory technicians in clinical laboratories is a particular challenge” (see revised manuscript, lines 68-71).

On line 21 the authors reference "slaughtered animals" as a source of these flukes but their paper proposes use of this technology in people. I recommend mentioning that adult flukes can come from humans as well. In the discussion it would also be good to mention that a potential short coming of this study is that it is based on flukes collected from animals only. It is not clear that the MALDI signal would be the same for flukes grown in humans. This limits the appropriate use of the new MALDI library to animal grown flukes until further validated is performed.

Response: We have included this suggestion as a limitation in the ‘Discussion’ (see revised manuscript, lines 301-304). However, owing to the fluke’s life cycle, we would expect that only eggs, but no adult worms, would be passed in the stools of infected human patients.

There was an inconsistency on line 282 that references "4 adult worms", this is not consistent with the rest of the text.

Response: For the in-house database development, we analyzed seven F. gigantica and one F. hepatica specimen. Next, we employed 3 additional F. hepatica and 75 additional F. gigantica samples for the external validation phase. If combining both study phases, a total of 82 F. gigantica and 4 F. hepatica specimens were subjected to MALDI-TOF MS. Hence, no corrective actions were taken.

In summary, the methods would not be easily reproducible based on this text alone. Additional information is needed before it is acceptable for publication. With the proposed revisions I can see this information being a useful contribution to the field.

Response: We are indebted to Reviewer #1 and very much hope that we were able to satisfactorily address all open issues.

Reviewer 2 Report

The manuscript microorganisms-1025147 describes the use of MALDI-TOF for the identification of adult Fasciola spp. from different origin. This manuscript is short and concise, it describes the employed techniques clearly and in a duplicable manner. The basis of this research is well-known, and it has been routinely employed for the identification of yeast and bacteria of clinical interest. Interestingly, few works related with Fasciola have been published with this technique, being mainly limited to identification of serum biomarkers during infection in sheep. A similar methodology has been previously described for the identification of cercariae, but Fasciola spp. has not among the studied organisms. Summarizing, this manuscript addresses an interesting topic with a well-known technique that can be a prove of concept for the development of MALDI-TOF peaks database for the identification of Fasciola spp. However, Reviewer must comment several aspects that author should revise before the consideration for publishing.

General comment

Nowadays PCR is a routinely technique for identification of a wide type of organisms. It is more common than MALDI-TOF and the technology is more reliable and precise. Authors must emphasize in MALDI-TOF is a faster and straight on technique, and often requires less sample and it can map tissues, if required.

Did authors found any difference among the samples conserved in ethanol and the ones stored in saline solution?

Author Response

Reviewer #2:

The manuscript microorganisms-1025147 describes the use of MALDI-TOF for the identification of adult Fasciola spp. from different origin. This manuscript is short and concise, it describes the employed techniques clearly and in a duplicable manner. The basis of this research is well-known, and it has been routinely employed for the identification of yeast and bacteria of clinical interest. Interestingly, few works related with Fasciola have been published with this technique, being mainly limited to identification of serum biomarkers during infection in sheep. A similar methodology has been previously described for the identification of cercariae, but Fasciola spp. has not among the studied organisms. Summarizing, this manuscript addresses an interesting topic with a well-known technique that can be a prove of concept for the development of MALDI-TOF peaks database for the identification of Fasciola spp. However, Reviewer must comment several aspects that author should revise before the consideration for publishing.

Response: We would like to express strong words of thanks to Reviewer #2 for the generous feedback and the careful reading of our manuscript.

General comment

Nowadays PCR is a routinely technique for identification of a wide type of organisms. It is more common than MALDI-TOF and the technology is more reliable and precise. Authors must emphasize in MALDI-TOF is a faster and straight on technique, and often requires less sample and it can map tissues, if required.
Response: We thank Reviewer #2 for this suggestion. We highlight the disadvantages of PCR-based techniques (e.g. the need for genus- or pathogen-specific primers) in the revised ‘Introduction’, where we state that PCR has “several limitations, including […] relatively high costs and unavailability of PCR-based testing using specific primers for e.g. trematodes outside highly specialized research laboratories. […] In contrast to PCR-based techniques, MALDI-TOF MS allows for a broad, ‘untargeted’ detection of microorganisms if these are present in the database used for pathogen identification” (see revised manuscript, lines 66-68 and lines 77-79).

Did authors found any difference among the samples conserved in ethanol and the ones stored in saline solution?

Response: We thank Reviewer #2 for this suggestion. In the current work, we did not comparatively assess the use of different storage solutions, which is acknowledged as a limitation, i.e. “Our study is limited by the small sample size for F. hepatica (i.e. only four adult worms), analysis of samples stemming from only two different geographical areas (i.e. Nigeria and Switzerland) and the use of samples preserved in different media (70% ethanol and 0.7% NaCl for F. gigantica and F. hepatica, respectively). It is important to mention that potential effects of different preservation media used were not thoroughly investigated in this proof-of-concept study” (see revised manuscript, lines 297-301).

***************************************************************************

Taken together, we thank the Editor and the two external reviewers for their interest in our work and the detailed comments and suggestions. We hope that our revised manuscript is now closer to the mark, and hence, might be considered for publication in microorganisms.

Round 2

Reviewer 1 Report

Thank you for addressing my concerns. I feel items have been appropriately addressed and the test is ready for publication.

While the text is appropriate, I want to share with the authors that adult flukes are occasionally removed from human livers and could benefit from this kind of identification. Contacting Clinical Microbiologists in endemic areas could be a source for material for future studies.

Author Response

Please find our detailed point-by-point response in the attached letter.
